# Perceptions of Diet Quality, Advice, and Dietary Interventions in Individuals with Diabetes-Related Foot Ulceration; A Qualitative Research Study

**DOI:** 10.3390/nu14122457

**Published:** 2022-06-14

**Authors:** Hailey Rae Donnelly, Clare Elizabeth Collins, Rebecca Haslam, Diane White, Peta Ellen Tehan

**Affiliations:** 1School of Health Sciences, College of Health Medicine and Wellbeing, University of Newcastle, Callaghan 2308, Australia; clare.collins@newcastle.edu.au (C.E.C.); rebecca.haslam@newcastle.edu.au (R.H.); peta.tehan@monash.edu (P.E.T.); 2Podiatry and Footcare Department, Hunter New England Local Health District, New Lambton Heights 2305, Australia; diane.white@health.nsw.gov.au; 3School of Clinical Sciences, Monash University, Clayton 3168, Australia

**Keywords:** diabetes-related foot ulceration, qualitative research, diabetes, nutrition, diet

## Abstract

Background: Dietary intake is a recognised contributor to healing in diabetes-related foot ulceration (DFU). However, it is currently unknown how individuals with DFU perceive their diet, and what is deemed an acceptable dietary intervention. Therefore, the aims of this study were to explore perceptions of diet quality, previous dietary advice, and dietary interventions in individuals with DFU, and secondly to determine acceptable dietary interventions in individuals with DFU to assist with wound healing. Methods: A qualitative study using reflexive thematic analysis was undertaken. Individuals with active or recent history of DFU were recruited from a high-risk foot service. Semi-structured interviews were undertaken. Results: Nineteen participants were included with three themes identified: A complex relationship with food, perceptions of food, diet and dietitians, and self-management. Dietary misconceptions were common. Self-perceived diet quality varied, with most unaware of how diet could impact wound healing. Many expressed barriers relating to food agency (purchasing, preparing, and accessing food). Participants expressed a strong preference for personalised, face-to-face dietary advice and nutritional supplementation. Conclusions: There is a need for personalised dietary re-education and assistance with food agency in this cohort to overcome commonly held misconceptions of diet and improve dietary intake to facilitate wound healing.

## 1. Introduction

Every hour in Australia, one person with diabetes has their foot or leg amputated [1]. Diabetes-related foot ulceration (DFU) and related amputations are estimated to cost the Australian healthcare system over AUS $600 million dollars annually [2]. In most cases, chronic DFU precedes amputation, with a range of factors implicated in reduced healing capacity [3]. Poor nutrition is an established contributor to delayed wound healing, with adequate intakes of energy, macronutrients and some micronutrients, including protein, zinc, vitamin C and vitamin D all deemed important for timely tissue repair [4,5,6,7]. In DFU populations, previous studies have demonstrated that poor diet quality, micronutrient deficiencies and malnutrition are highly prevalent [8,9,10,11,12], with one study demonstrating up to 62% of DFU patients are malnourished [12]. Therefore, optimising dietary intake may improve wound healing in DFU, but also improve blood glucose control, weight management and cardiovascular risk factors in this highly vulnerable population [13,14].

Self-perceived diet quality has not been explored in individuals with DFU, but it has been investigated in the broader population with diabetes. A previous study demonstrated that misconceptions about dietary intake are common amongst individuals with diabetes, particularly in relation to specific food groups such as fruit and dairy [15]. Previous research has suggested that behavioural beliefs and attitudes influence dietary behaviours [16,17]. In a previous case-control study [18] and pilot study [19], nutrition education was shown to be effective in improving nutrition knowledge in those living with type 2 diabetes [18,19]. However, more research is needed to determine how interventions could be modified to increase knowledge, improve dietary intake and facilitate long-term behaviour change.

To optimise dietary intake, a number of dietary interventions have been reported, including nutrient supplementation and education [8,20,21,22,23,24,25,26,27]. However, evidence around efficacy of dietary interventions in individuals with DFU is conflicting and generally of variable scientific quality [8,20,21,22,23,24,25,26,27]. A recent randomised controlled trial demonstrated nutritional supplementation in combination with nutrition education improved wound healing outcomes in patients with DFU, with a mean decrease in wound area (mm^2^/w) [22]. A systematic review investigating nutritional supplementation of a range of nutrients for enhancement of healing in DFU, including zinc, magnesium, omega-3, vitamin D and probiotics, found apparent benefit with all five nutrient supplements in regards to secondary outcomes of wound depth, width and length [26]. Some common methodological limitations within previously published research includes unclear definitions of standard care [26], or not measuring participant adherence to the dietary intervention [23,26]. Furthermore, there has been no investigation into patient acceptability of these interventions.

When designing and implementing a dietary intervention that is both effective and acceptable, and maintains high levels of engagement, adherence and long-term behaviour change, perspectives of individuals with DFU need to be taken into consideration [28]. Participants’ perspectives on which nutrition interventions are acceptable have rarely been considered prior to designing an intervention, with researchers relying on professional insight, which lacks lived experience [29]. Clinicians and researchers can have individual bias towards interventions they believe are appropriate without considering patient perspectives [29]. Patient perspectives can help define new interventions to be explored from personal experience, allowing relevant and quality research to be formed away from dominating dietary interventions already being studied with limited efficacy [29]. Patient perspectives can also allow for new insights into difficulties or issues with existing interventions [29]. Collaborative co-design considers the patient perspective, and has been shown to increase participant engagement and long-term sustainability of interventions [30].

Therefore, the primary aim of this study was to explore perceptions of diet quality, previous dietary advice, and dietary interventions in individuals with DFU, and secondly to determine acceptable dietary interventions in individuals with DFU to assist with wound healing.

## 2. Materials and Methods

### 2.1. Qualitative Approach & Research Paradigm

This was a qualitative research study using a reflexive thematic approach of semi-structured interviews. This methodology was utilised to explore the perspectives of individuals with DFU on their dietary intake and dietitians, as well as their perceptions on various dietary interventions. This approach allows for generation of new insights, meanings, and concepts to be identified to achieve study aims of exploring the perceptions of those with DFU.

### 2.2. Researcher Characteristics and Reflexivity

One researcher recruited and screened participants for eligibility (DW). Two researchers (PET & DW) had a clinician-patient relationship with a small number of participants. Researchers involved were clinicians including two podiatrists and two dietitians experienced in diabetes management, and one researcher (HRD) was a final year Nutrition and Dietetics Honours student. Both researchers (HRD & PET) who were involved in data analysis maximised the utility of their varied health professional backgrounds and perspectives during the coding and theme creation process.

### 2.3. Context

Participants were recruited from a high-risk clinical foot service in Newcastle, NSW, Australia. Eligibility for this publicly funded service includes an active foot ulceration or acute foot presentation related to a diabetes diagnosis. The service sees approximately 120 patients per week, is staffed by five podiatrists, an allied health assistant, and facilitates a multi-disciplinary clinic one day per week with orthopaedics, endocrinology, and diabetes education.

### 2.4. Sampling Strategy

Inclusion criteria for the current study included those 18 years and over with diabetes mellitus (type 1 or 2), with active foot ulceration or a recent history of foot ulceration (within last 3 months), ability to speak English fluently and an ability to provide written informed consent. Individuals were not eligible for the study if they had a history of cognitive disorders impacting their ability to give consent or communicate effectively. A targeted sampling strategy was used to capture a range of gender, age, body mass index, living situations, education level and diabetes type. Potential participants were provided an invitation with their appointment for their next visit to the clinic by researcher (DW) and those who expressed interest were provided an information statement and subsequently offered an interview session.

### 2.5. Ethical Issues Pertaining to Human Subjects

Prior to enrolment, eligible participants provided written informed consent. The study was approved by the Hunter New England Local Health District Human Research Ethics (LNR 2020/ETH 02845 and SSA 2020/STE05004).

### 2.6. Data Collection and Instruments

Participants attended a single study visit between January and April 2021. Interviews took place in a clinical room, to ensure privacy and confidentiality. All interviews were recorded on a digital device and conducted by one researcher (HRD) supervised by a clinical podiatrist (PET). It was disclosed to participants that HRD was a final year dietetics student. Interviews were designed to take twenty minutes, depending on participant engagement. Interview questions were developed following a literature review of research exploring dietary interventions for individuals with ulceration, and peer-reviewed within the research team. Based on a content analysis of retrieved literature, seven questions and an interview script were then derived (Appendix A). Questions were a combination of open-discovery and closed questions, exploring participants’ perceptions of diet quality, if they felt improvements could be made within their diet and if they would be open to dietary assistance. Participants were also asked about their previous experience with dietitians and if they were aware of the relationship between nutrition and wound healing. Interviewers proposed numerous types of dietary interventions to participants who were asked to provide their opinion on each and their preferences. The opening question, “If you wouldn’t mind telling me a little about your foot ulcer?” was included to create rapport between participant and interviewer before addressing the main questions.

### 2.7. Clinical Measures

Basic demographic questions relating to participants’ age, gender, diabetes type and duration, and foot ulcer duration were collected. Participants’ mean Index of Relative Socio-economic Disadvantage (IRSD) was determined by participants’ post code [31]. The IRSD is a broad measure of disadvantage that combines the economic and social situations of people within an area [31]. Furthermore, participant’s weight and height were measured by two researchers (PET and HRD) collaboratively using scales and a stadiometer, recorded to the nearest whole number. These clinical measures were collected to provide adequate sample description and to determine heterogeneity.

### 2.8. Units of Study

The sample size of nineteen participants was determined throughout the interview process as researchers felt data obtained was rich enough to identify shared and meaningful patterns [32]. Data saturation was agreed upon by two researchers when no new themes were able to be derived and felt no further data collection or coding was necessary to provide value to the study. Duration of interviews ranged from five to 22 min, with a median duration of 13 min.

### 2.9. Data Processing

Interview data were transcribed verbatim by a single researcher (HRD). Maintaining the interviews’ original nature when transcribing is embedded in the reflexive thematic approach [33]. Transcribed interviews were then imported into NVivo^®^ software (QSR International ©, Melbourne, Australia).

### 2.10. Data Analysis

Two researchers (HRD and PT) utilised Braun and Clarke’s six phases of thematic analysis to analyse the dataset [33]. Both researchers read through the dataset multiple times to become immersed in the data, aligning with the reflexive thematic approach [33]. Through this intimate familiarisation, patterns were identified, with semantic codes generated through labelling important features that had potential to be relevant to answering the research aim. Once both researchers had independently coded the interviews, the similarities and differences of the codes were discussed to increase the level of engaement of both researches with the data. Broader patterns of meaning through collation and clustering of codes enabled the development of potential initial themes. Themes were refined, and compared to the dataset, to ensure they reflected the data, and that they answered the research questions. Final themes (patterns of meaning) were then systematically identified, organised, and named to provide insight across the entire dataset (PET and HRD) [34]. This allowed understanding of participants’ lived experience with DFU. All researchers provided insight and agreed to final coding and development of themes.

### 2.11. Techniques to Enhance Trustworthiness

To enhance trustworthiness and credibility of the data analysis, peer debriefing was conducted with the research team. One researcher (PT) further debriefed with podiatrists working in the high-risk foot service by attending a meeting, presenting drafted themes, and gathering feedback.

## 3. Results

### 3.1. Participant Characteristics

Nineteen participants were included in the study, with a mean age of 66 years (standard deviation (SD) 10). With 68% identifying as male, this was reflective of the DFU population at the high-risk foot clinic, with a previous study showing DFU patients being predominantly male [35]. The mean BMI for participants was 30 kg/m^2^ (SD 5) with half (52%) of participants classed as overweight or obese based on their age [36]. The mean IRSD score was 965.1, indicating all participants were relatively disadvantaged [37]. Participant characteristics are presented in Table 1.

### 3.2. Themes

Three themes were generated, as follows: (1) a complex relationship with food; (2) perceptions of food, diet and dietitians; and (3) self-management.

#### 3.2.1. A Complex Relationship with Food

Participants discussed their relationship with food with a sense of great emotion, ranging from a love of food to a hate of the rules that impact their freedom to make choices. Many participants expressed a sense of mourning over the times before their diabetes diagnosis, when they could choose to eat freely, and that their disease had taken this freedom from them.


*“Not having diabetes [would help me eat better]. That would be fantastic because I really love my food. There is not one food I don’t like”.*

*Participant 12, male, 60 years*


Many of the participants referred to the potential consequences of their food choices and discussed the difficulties of complying with what they were supposedly allowed to eat. Some expressed a negative relationship with food, emphasising their struggle with portion control, as well as difficulties with self-control around snack and discretionary foods. Some participants further felt the need to justify their discretionary food intake, citing their blood glucose control as permission to indulge. Many participants further discussed feelings of surprise relating to how significant their food intake was on their overall health and that this was an unexpected part of their diabetes management.

The preparation of food further elicited a range of emotions, with many seeing this as a joyful daily event that was a central part of day-to-day life. Many participants spoke positively about going to the supermarket to procure food, then going home to cook a meal for themselves and their families.


*“I mean at the end of the day, as a diabetic it’s something that is a big part of your life, is diet”.*

*Participant 9, male, 45 years*


Some participants also suggested that both cooking and preparing a meal provided them with a sense of purpose and were subsequently resistant to the concept of receiving assistance in this area, as it was perceived as challenging their independence.


*“I don’t want [to] lose our ability to do my own cooking. I can cook, I like to cook”.*

*Participant 16, male, 66 years*


Whilst some felt cooking and preparing food was an enjoyable event, others felt it was a chore, and had negative attitudes towards cooking. Food preparation was seen as a burden by some, with multiple barriers cited to enjoyment. One barrier was the perception that fresh food and vegetables were expensive, whilst others were dependent on family members to grocery shop for them due to their limited mobility as a consequence of their DFU. Some participants also expressed variety exhaustion arising from consuming the same meals frequently.


*“When I done the education thing, the diabetes, they said you can have as much salad as you like, [but] you can’t eat that much of it cause you do get sick of just [salad]”.*

*Participant 14, male, 65 years*


There were some gender-based differences in attitudes towards preparation of food. Most of the female participants discussed that cooking was part of their role in the household and was subsequently part of their identity. When suggestions of assistance were made in relation to cooking, this was seen as undermining their ability to cook, and their matriarchal role in the household. Many male participants who lived with a spouse on the other hand, expressed general disinterest in cooking, and suggested that this was the woman’s role within the household.


*“No, its [wife’s name] job. She cooks just about every night”.*

*Participant 17, male, 57 years*


#### 3.2.2. Perceptions of Food, Diet and Dietitians

Most participants had seen a dietitian at the time of their diabetes diagnosis, which for most was over twenty years prior, with many not engaging at all since that time. Several different diet types were discussed, describing either what they were currently following or had followed in the past, including Pritikin, Mediterranean, vegetarian and low-fat diets, with most aiming to achieve weight loss. Participants commonly expressed misconceptions around food, including what foods they were supposedly not allowed to consume and what foods needed to be consumed with caution. These foods frequently included fruit and dairy.


*“I know dairy is deadly for a diabetic, like ice-creams and yoghurts and things”.*

*Participant 12, male, 60 years*


Whilst the source of these misconceptions is unknown, participants discussed obtaining dietary advice not only from dietitians, but also from other health professionals including primary health care physicians and registered nurses. External sources of dietary information including the internet were also utilised by participants.


*“There’s nothing I haven’t seen on Google”.*

*Participant 12, male, 60 years*


Participants’ self-perceptions of dietary intake were varied. Some participants described their diet as poor, whilst others mentioned there was no room for improvement or rather, it was perfect. A common perception was that participants felt they knew how to eat healthily, but acknowledged they needed to make changes and appeared open to recommendations. Experiences with dietitians were also varied. Some felt dietitians were a great source of helpful information relating to their diabetes management, with one participant describing a previous positive, personalised approach. However, negative experiences were far more common, with some viewing dietitians as an authoritative figure who would discipline them for making poor choices. Some participants also expressed that advice given was not tailored to their personal needs, but rather a one-size-fits-all approach. Many participants also perceived engagement with a dietitian as unnecessary, as they felt they knew everything already, or utilised external sources of information. Furthermore, some participants believed seeing a dietitian would not provide any additional value to their diabetes management, as they felt their first and only appointment at diagnosis was enough.


*“I’ve been there, done that. They tell me everything I need to know, and I already know”.*

*Participant 18, female, 65 years*


Conflicting information from dietitians over time was very common and led to a sense of confusion and subsequent resignation. This often resulted in participants to question the reliability of information and education provided to them.


*“I don’t know, I came out confused, because things I thought I was told I could eat she said no, and the things I thought I couldn’t have, she said yes definitely. Then I went to another one not long after and it was all different again, and I just got very confused”.*

*Participant 11, female, 65 years*


In relation to dietary intake and wound healing, none of the participants could recall being given specific advice related to improving wound healing. Many expressed surprise that diet could impact their wound healing capacity. Whilst many participants were curious, others expressed that they lacked hope in the possibility that dietary change could benefit their wound healing.


*“I don’t think diet is going to fix that unfortunately”.*

*Participant 9, male, 45 years*


#### 3.2.3. Self-Management

Participants individual wound journeys were all unique. However, all of the stories were told with a sense of trauma, apprehension, caution and general fear of amputation. Many also discussed that their DFU occurred suddenly and weren’t aware of the severity or that they were at risk of ulceration or amputation at all. Numerous participants expressed mistrust in health professionals outside of the high-risk foot clinic and their ability to help heal their wounds, with some suggesting their wounds were previously *“wrongly looked after”, Participant 3, male, 60 years*. Some participants held negative perceptions of medical professionals as disciplinarians who did not listen, rather than supportive caring professionals. The idea of physicians preferring to amputate rather than considering other treatments first was also expressed by some participants. Some participants also had their own opinions on wound healing therapies with some trying alternative treatments. Due to this constant struggle with self-managing their DFU, participants were open to having any assistance that could expedite the wound healing process.

Through discussion with participants in interviews, it was evident which dietary interventions participants found potentially acceptable. Supplementation to improve wound healing was viewed positively by most, with many expressing a sense of desperation to heal their wound and a willingness to try a dietary intervention if it had potential to help.


*“If it would help with the wound healing, you’d have to take it, you’d be silly not to!”.*

*Participant 14, male, 65 years*


Whilst most were interested in supplementation, some participants expressed concern of the interaction of a new medication with their current medication regime, and some felt they were already taking too many tablets.


*“I just don’t like pumping pills into me”.*

*Participant 18, female, 65 years*


Most participants described the desire for dietary interventions to be delivered in a person-centred, individualised fashion, with participants showing preference for face-to-face delivery rather than telehealth. Most participants disliked telehealth due to a lack of technological ability or equipment. Many also enjoyed the event of coming into the clinic as a form of social interaction as many participants lived alone. However, some participants were happy to see a dietitian via telehealth as they felt it would save time.

Whilst participants expressed a willingness to trial nutrition interventions that may assist with their wound healing, they expressed distinct preferences. Although personalised dietary interventions achieved a positive response from participants, when asked about group workshops, cooking classes, recipe ideas, precooked meals and meal kits, these were generally deemed as not acceptable. The majority of participants felt group workshops were impersonal and many disliked the idea of cooking classes/meal planning. Whilst the majority were not interested in group sessions, a small minority saw it as an opportunity to socialise and learn something new. Healthy recipe ideas were not acceptable to participants who had a general dislike of cooking, and for those who enjoyed cooking, they felt they didn’t need assistance. Precooked meals were not viewed favourably as they were seen as unsustainable due to the financial outlay.


*“They’re so expensive… for what you spend a week, I can make that cover a fortnight”.*

*Participant 18, female, 65 years*


## 4. Discussion

The results of the current study revealed perceptions of current dietary intake, previous dietary interventions, and determined acceptability of future dietary interventions in individuals with DFU. Generated themes demonstrated that factors influencing dietary intake were complex and multi-faceted.

The first theme, ‘A complex relationship with food’, explored participants varied relationships with food and cooking, with a sense of grief and loss commonly expressed over the loss of freedom of choice associated with their diabetes diagnosis, which for many was over twenty years ago. This concept of strict rules relating to what they were allowed to eat combined with outdated dietary information contributed to an unhealthy perception of food. The second theme, perceptions of food, diet and dietitians, described the variety of beliefs and commonly held misconceptions participants had of food, nutrition, and diet, and participants varied experiences with dietitians. The final theme, ‘self-management’, highlights participants’ desire to reduce the burden associated with living with a chronic ulceration.

Participants felt they had a lack of freedom of choice, and that their dietary intake was determined by their insulin dosage which was fixed. Historically, food intake was set to an insulin dosage, and individuals were told not to adjust [38]. However, in the contemporary context, diabetes education has shifted away from set food intake and insulin dosages to allow for greater flexibility and freedom of food choice in line with the increased availability of insulin products with variable profiles [38]. A previous qualitative study in a diabetes cohort reported similar difficult personal relationships with food [39]. This related to challenges accessing healthy food and implementing dietary change, as this was a large shift from their dietary norms [39]. Therefore, nutrition education for this population should include addressing this misconception of set insulin dosages in order to increase food choice freedom and to potentially improve this population’s relationship with food.

In relation to cooking, there was dissonance between participants’ confidence in their cooking skills and their concerns and barriers relating to cooking healthy meals. Participants expressed capability with cooking and declined assistance with food agency (ability to obtain and prepare food within their social, physical, and economic environment [40]). However, they complained about the cost of fresh fruit and vegetables, physically being able to shop for their food, their struggle for independent cooking and lack of meal variety, most of which suggest low levels of food agency [40]. In relation to physical abilities, individuals with active DFU wear cumbersome offloading devices which impacts on their mobility, in addition to being told to reduce their weightbearing activity [41]. This impacts on their independence and subsequent ability to procure ingredients and food, with many needing to rely upon others. However, with increased food agency skills, this population could potentially reduce their consumption of discretionary and processed foods and increase their ability to prepare more nutritious homemade meals [40]. Improved food agency could also assist with the financial strain of purchasing fresh food which is in season at lower cost which many cited as a barrier to eating well. Furthermore, traditional gender roles and a sense of identity, and independence associated with cooking [42], may have contributed to limited self-awareness that they could benefit from assistance. Previous research exploring the effectiveness of gender-specific dietary interventions compared to gender-neutral interventions determined that a larger proportion of gender-specific interventions were effective in improving nutrition [43]. This highlights the need for both food agency and gender to be considered when designing future dietary interventions in this cohort.

Discussion of dietary intake and diet types with participants revealed frequent and varied dietary misconceptions. One of the most common misconceptions related to intake of fruit and dairy, with participants expressing caution due to the carbohydrate and sugar content. However, these foods are nutrient rich and can contribute to diet quality and optimise wound healing capacity [4,6,7]. Previous qualitative research in people with type 2 diabetes described similar misconceptions, with fruit and dairy perceived as ‘bad’ foods [15], and two questionnaires completed by people with type 2 diabetes suggested their attitudes towards foods being “good” or “bad” influences their dietary behaviours [16,17]. This further emphasises the need for dietary education in this population. Whilst participants generally viewed their diet positively, further questioning and discussion revealed potential deficits. Previous research has demonstrated that dietary quality in this population does not meet recommended guidelines, lacking essential nutrients for wound healing, including inadequate protein and folate intake [35]. Other studies in DFU populations have also determined that micronutrient deficiencies such as vitamin D, C, A, and zinc are common [10]. These nutrients are required to regulate synthesis of collagen, and extracellular matrix formation, essential to wound healing [10,44]. This further justifies the need for regular dietary assessment and intervention to optimise nutrition and accelerate healing capacity [45].

Most participants had not sought dietary advice since their diagnosis, which was up to 20 years prior, and believed dietary advice had not evolved and were therefore unaware of more current nutrition evidence and dietary recommendations. Whilst most participants in the current study had received dietary advice at diagnosis, contrary to current findings, a recent qualitative study found many participants with diabetes had not received any dietary advice from a health professional and for those that had received dietary advice, it was in the form of a generic healthy eating pamphlet [46]. Participants in the current study felt that the dietary advice they had received was conflicting, and inconsistencies arose not just between dietitians, but also other health professionals. This finding is consistent with a previous study [46], with a lack of and conflicting dietary advice likely contributing to confusion and sub-optimal dietary choices in this population. All participants also expressed they had never been given dietary advice in relation to wound healing and lacked understanding of the importance of this relationship. This is likely related to limited engagement with dietitians since their diagnosis and no consultation since developing a DFU. Previous studies in patients with pressure ulceration demonstrated a positive association between wound healing and individualised nutrition care by a dietitian in combination with nutritional supplementation specific for wound healing [47]. Individualised nutrition care specific to healing could, therefore, be useful in this population.

Most participants were open to dietary intervention if it had the potential to help their wound heal. This motivation to make dietary change due to the complications associated with diabetes was also identified in a recent qualitative study exploring individuals recently diagnosed with type 2 diabetes [48]. Although participants were open to interventions that may assist with wound healing, there was an underlying scepticism towards healthcare professionals with many expressing a lack of trust in the broader medical profession. A recent qualitative paper seeking to understand people’s experiences living with Charcot neuroarthropathy reported mistrust in healthcare professionals, believed they were misdiagnosed, and blamed them for their chronic condition [49]. The idea of medical professionals as disciplinarians was also confirmed in a recent qualitative study where one participant described medics as dictators [46]. Some participants in the current study also questioned physicians’ motives, suggesting that some medical professionals would prefer to amputate rather than give the wound a chance to heal. These feeling of mistrust and scepticism towards the medical profession favouring amputation has also been reported by a previous qualitative study in individuals with diabetes with a history of foot ulceration and/or minor amputation [39]. Therefore, future educational programs will need to consider first building trust with this population prior to implementing an intervention.

People with DFU are seeking a dietary intervention that is personalised, aimed at achieving wound healing, and implemented by health professionals that are respectful, supportive and trustworthy. Personalised advice from a dietitian has also previously been cited as preferable in diabetes populations [46]. Individuals with DFU also desire nutrition supplementation to support their wound healing, and more confidence in decision making around food to give a returned sense of freedom from rules around intake. The current findings indicate a personalised intervention that targets their specific barriers to change, such as skills in food agency, may be more appropriate for individuals with DFU.

### Limitations

Limitations in the current study need to be acknowledged. Whilst we made every effort to recruit a heterogenous sample, the population may not be generalisable to the broader Australian or international DFU population. A targeted sampling strategy was used in an attempt to achieve a heterogenous sample, in particular age, gender, living situation and education level. Another limitation was the exclusion of individuals who did not speak English, and ethnicity information was not collected. Therefore, it is not possible to determine if the sample represents a range of cultural groups or is reflective of the broader multi-ethnic population.

## 5. Conclusions

The current study demonstrates that those living with DFU may benefit from assistance with improving their food agency and dietary patterns, and that participants were open to dietary interventions if the focus was on expediting healing of their wounds. Future studies addressing dietary intake with the aim of DFU healing should provide personalised, gender-specific dietetic support that takes into consideration the individual’s barriers to change. Medical nutrition therapy interventions should be provided face-to-face and include personalised nutritional supplementation to complement the dietetic support. Dietary behaviour change could potentially have a positive impact on this population’s relationship with food, overall diet quality, and subsequently may improve wound healing outcomes.

## Figures and Tables

**Table 1 nutrients-14-02457-t001:** Descriptive characteristics of nineteen diabetes-related foot ulceration patients participating in semi-structured interviews.

		Range
Participants (*n*)	19	
Mean Age, *n* (SD ^a^)	66 (10)	45–90
Male gender, *n* (%)	13 (68)	
Living situation, *n* (%)		
Alone	9 (47)	
With partner	10 (53)	
Education Level, *n* (%)		
Did not complete high school	6 (32)	
Completed high School	3 (16)	
Trade Certificate	8 (42)	
University Degree	2 (11)	
Mean Height (cm ^b^)	175	162–189
Mean Weight (kg)	94	70–125
Mean BMI ^c^ (kg/m^2^) ^d,e^	30.8	23.2–43.1
BMI Class, *n* (%)		
Healthy Weight	9 (47)	
Overweight	6 (32)	
Obese	4 (21)	
Diabetes, *n* (%)		
Type 1	4 (21)	
Type 2 (requiring insulin)	8 (42)	
Type 2 (not requiring insulin)	7 (37)	
Mean Diabetes duration, years (SD)	21 (8)	7–38
Mean HbA1c ^f^ (%)	7.2	5.8–11
Active foot ulceration (*n*)	15	
Previous amputation, *n* (%)	8 (42.1%)	
Previously received nutrition advice, *n* (%)	16 (84%)	
Previously received diabetes education, *n* (%)	17 (89.5%)	
Mean Index of relative socio-economic disadvantage	965.1	660–1075

^a^ SD = standard deviation, ^b^ cm = centimetres, ^c^ BMI = body mass index, ^d^ kg = kilograms, ^e^ m = metres, ^f^ HbA1C = glycosylated haemoglobin. Index of socio-economic disadvantage = (defined by the Australian Bureau of Statistics). Body Mass Index Class: Healthy weight range < 65 years (18.5–24.9 kg/m^2^); Overweight (pre-obese) < 65 years (25.0–29.9 kg/m^2^); Obese < 65 years (≥30 kg/m^2^); Healthy weight range > 65 years (24–30 kg/m^2^); Overweight years > 65 years (≥30 kg/m^2^) [36].

## Data Availability

Due to ethics, restrictions data is not available.

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
