# Peer review of "Perceptions of Diet Quality, Advice, and Dietary Interventions in Individuals with Diabetes-Related Foot Ulceration; A Qualitative Research Study"

_nutrients, 2022, doi:10.3390/nu14122457_

Round 1

Reviewer 1 Report

I would like to thank the authors for their work. Their results were very nicely presented and provided nice insight into the breadth and depth of the participant responses. I have only a few notes that will help clarify some of the methodologies and minor editorial comments. 

1. Line 49 -- I believe there is an excess period and the word "and" is missing.

2. Line 101 -- This reads very textbook. Improve by making more specific to the study. 

3. How was the coding process done? Did 1 or 2 researchers code the interviews? If 2, did both code all interviews? How were disagreements dealt with?

4. Table 1 -- please add % to each of the cells that currently only report n. 

5. I believe a figure displaying which codes were used to populate each of the themes would be beneficial. 

6. Only the description of the first theme provided any mention of comparisons. Were any other comparisons examined? Were there no differences for any of the other themes noted? Please note in the methods which comparisons were examined and in the results discuss whether differences were observed by any of those comparisons. 

Author Response

Reviewer 1

I would like to thank the authors for their work. Their results were very nicely presented and provided nice insight into the breadth and depth of the participant responses. I have only a few notes that will help clarify some of the methodologies and minor editorial comments. 

1.Line 49 -- I believe there is an excess period and the word "and" is missing.

Thank you for identifying this error, this has now been amended.

  1. Line 101 -- This reads very textbook. Improve by making more specific to the study. 

Thank you for this comment, we have altered this based on your feedback

Of exploring the perceptions of those with DFU” has been added to the sentence, “This approach allows for generation of new insights, meanings, and concepts to be identified to achieve study aims of exploring the perceptions of those with DFU” to make it more specific to the study.

  1. How was the coding process done? Did 1 or 2 researchers code the interviews? If 2, did both code all interviews? How were disagreements dealt with?

Thank you for this important query, yes two researchers coded the interviews. Part of the reflexive thematic analysis method is the pooling of codes and interpreting them as a shared resource bank, rather than looking for agreement, on which to build themes [1]. The reflexive thematic analysis method supports different perspectives being bought to the data analysis. Therefore we did not seek consensus, but rather pooled and built upon the different codes.

Under 2.10 Data Analysis on line 164 it says, “Two researchers (HD and PT) utilised Braun and Clarke’s six phases of thematic analysis to analyses the dataset”. 

We have added the following on line 169: Once both researchers had independently coded the interviews, the similarities and differences of the codes was discussed to increase the level of engaement of both researches with the data.

  1. Table 1 -- please add % to each of the cells that currently only report n. 

Thank you this have been amended to include %

  1. I believe a figure displaying which codes were used to populate each of the themes would be beneficial. 

Thank you we appreciate this comment however reflexive thematic analysis does not require population of codes to create themes and we do not believe that presenting the codes in a figure would be consistent with the reflexive thematic approach.  Rather the theme should capture something important in relation to the research question. The ‘keyness’ of a theme is not necessarily dependent on quantifiable measures - but rather on whether it captures something important in relation to the overall research question [2]. Further, consistent with the qualitative approach, there is no hard and fast answer to the question of what proportion of your data set needs to display evidence of the theme for it to be considered a theme [2].

  1. Only the description of the first theme provided any mention of comparisons. Were any other comparisons examined? Were there no differences for any of the other themes noted? Please note in the methods which comparisons were examined and in the results discuss whether differences were observed by any of those comparisons. 

Thank you for this comment. Yes the first theme contained a few different comparisons, between some participants having one way of thinking, and others something different. There were also comparisons within participants themselves where they compared their diet and lifestyle prior to their diabetes diagnosis. The other themes did not really elicit the same comparisons.

  1. Terry G, Hayfield N. Essentials of Thematic Analysis. Washington, DC 20002: American Psychological Association; 2021 31/05/2022.
  2. Braun V, Clarke V. Using thematic analysis in psychology. Qualitative Research in Psychology. 2006;3(2):77-101.
  3. Collins R, Burrows T, Donnelly H, Tehan PE. Macronutrient and micronutrient intake of individuals with diabetic foot ulceration: A short report. J Hum Nutr Diet. 2021.
  4. Armstrong DG MJ, Molina M, Molnar JA. Nutrition interventions in adults with diabetic foot ulcers. Guideline Central. 2021.

5. Terry G, Hayfield N, Clarke V, Braun V. Thematic Analysis. In: Willig C, Stainton RW, editors. The sage handbook of qualitative research in psychology SAGE Publications; 2017. p. 18.

Reviewer 2 Report

The paper brings up a question that how diet related to DFU condition and the healing process. It is joyful to reading through this paper because it quotes a lot real conversation and readers could have emotional connections to the participants, however, this also makes the paper more similar to a report on new paper rather than a academic publication. There are some suggestion:

1. In results 3.2 themes, under each themes, author described different situation in different patients as "some patients", "many participants". It will be better to have the actually percentage of patients in each opinion (or choice/situation).

2. In abstract, author mentioned DFU is associated with malnutrition. In this study, 50% of the patients are overweighted, does these participant also in a malnutrition condition?

3. What's the diet component distribution in these participant? How these different from a ideal diet?

Author Response

Reviewer 2

The paper brings up a question that how diet related to DFU condition and the healing process. It is joyful to reading through this paper because it quotes a lot real conversation and readers could have emotional connections to the participants, however, this also makes the paper more similar to a report on new paper rather than a academic publication. There are some suggestion:

Thank you for this kind comment and we are glad that you enjoyed reading our work. The conversational narrative style of writing and use of quotes is consistent with the qualitative approach which we have utilised as an academic style.

  • In results 3.2 themes, under each themes, author described different situation in different patients as "some patients", "many participants". It will be better to have the actually percentage of patients in each opinion (or choice/situation).

Thank you for this comment. We appreciate that if utilising a quantitative approach this would be seen as imprecise. However, with reflexive thematic analysis in qualitative research, the quantity of each opinion is not viewed as important but rather, that the theme captures something important in relation to the research question. There is no hard and fast answer to the question of what proportion of the data needs to display evidence to the theme. Researcher judgement is necessary to determine what a theme is, and that this is flexible. The keyness of a theme is not necessarily dependent on quantifiable measures - but rather whether it captures something important in relation to the overall research question [2].

  • In abstract, author mentioned DFU is associated with malnutrition. In this study, 50% of the patients are overweighted, does these participant also in a malnutrition condition?

Thank you for this comment, we agree it would be interesting to assess the malnutrition prevalence in this population, and we plan on exploring this in the future. Our aim was to explore perceptions, rather than actually measure dietary intake or screen for malnutrition. Whilst it is highly likely the participants diets have macronutrient and micronutrient deficiencies and we have measured nutrient intake in this cohort previously [3], this was not the aim of the present study. 

  1. What's the diet component distribution in these participant? How these different from a ideal diet?

Thank you for this comment and again, we agree that this would be interesting, however in this qualitative study, we investigated individuals’ perceptions of diet quality, dietary advice, and dietary interventions. Whilst it is highly likely the participants diets have macronutrient and micronutrient deficiencies and we have measured nutrient intake in this cohort previously [3], this was not the aim of the present study. 

We agree it would be interesting to compare dietary perceptions and actual dietary intake of those living with diabetes-related foot ulceration, as the nutritional requirements for this population differs from those living with diabetes without a foot ulceration and other wound aetiologies [4]. We plan on exploring this in the future.

  1. Terry G, Hayfield N. Essentials of Thematic Analysis. Washington, DC 20002: American Psychological Association; 2021 31/05/2022.
  2. Braun V, Clarke V. Using thematic analysis in psychology. Qualitative Research in Psychology. 2006;3(2):77-101.
  3. Collins R, Burrows T, Donnelly H, Tehan PE. Macronutrient and micronutrient intake of individuals with diabetic foot ulceration: A short report. J Hum Nutr Diet. 2021.
  4. Armstrong DG MJ, Molina M, Molnar JA. Nutrition interventions in adults with diabetic foot ulcers. Guideline Central. 2021.

5. Terry G, Hayfield N, Clarke V, Braun V. Thematic Analysis. In: Willig C, Stainton RW, editors. The sage handbook of qualitative research in psychology SAGE Publications; 2017. p. 18.

Reviewer 3 Report

The manuscript describes the results of a qualitative research study investigating the perceptions of diet quality and dietary interventions in people suffering diabetes-related foot ulceration, with the aim of determining dietary interventions able to favour wound healing. The study has been carried out using semi-structured interview.

As also underlined in the Limitation section, the main critical point of the study is the number of patients enrolled. The authors affirm that the sample size "was determined throughout the interview process as researchers felt data obtained was rich enough to identify shared and meaningful patterns", but, clearly, using this method the number of enrolled patients is not enough to produce results that can be generalized.

Moreover, the authors declare that the sampling strategy allows to obtain a heterogeneous sample, but no information about these different characteristics of the patients is reported.

The Results section is presented in too narrative form. The sentences: "Many of participants referred…." or "Some expressed" must be avoided since no information about the numerical relevance of the phenomena is provided. For the same reason, the individual statements have to be eliminated.

A table reporting the percentage distribution of the answers could help in analyzing the main results of the interview.

The Introduction is too long and several concepts and comments are present in both Introduction and Discussion (lines 52-55 and 392-397), or discussed in the Results.

Even if the patients number is very small, the correlation between the perception of food and some parameters (for example BMI class, or Diabetes type or duration) should be evaluated, this improving the impact of the study.

Author Response

Reviewer 3 

The manuscript describes the results of a qualitative research study investigating the perceptions of diet quality and dietary interventions in people suffering diabetes-related foot ulceration, with the aim of determining dietary interventions able to favour wound healing. The study has been carried out using semi-structured interview.

As also underlined in the Limitation section, the main critical point of the study is the number of patients enrolled. The authors affirm that the sample size "was determined throughout the interview process as researchers felt data obtained was rich enough to identify shared and meaningful patterns", but, clearly, using this method the number of enrolled patients is not enough to produce results that can be generalized.

Moreover, the authors declare that the sampling strategy allows to obtain a heterogeneous sample, but no information about these different characteristics of the patients is reported.

Thank you for these considered comments. We have utilised a qualitative method, and as part of this method, the sample size is determined based on the richness of the data, rather than on the number of participants. According to the Sage handbook of qualitative Research, “The most important aspect of data type or mode of collection is quality of the data. Rich and complex data on a given topic are the crown jewels of qualitative research, allowing us deep and nuanced insights. Quantity (e.g., sample size) is also a consideration, but should not be conflated with quality. Key in thinking about sample size in TA is to recognise that it produces accounts of patterns across the dataset” [5].Thematic analysis is not about the number of participants but the richness of the data and it’s ability to create meaningful information [5]. 

Our heterogenous sample is outlines in table 1, and as you can see there is a range of age, gender, employment status, education, weight and previous amputation. However we agree that whilst we utilised a heterogenous purposeful sampling method, our findings may not be generalizable and have added the following to the limitations section.

Limitations in the current study need to be acknowledged. Whilst we made every effort to recruit a heterogenous sample, the population may not be generalisable to the broader Australian or international DFU population. A targeted sampling strategy was used in an attempt to achieve a heterogenous sample, in particular age, gender, living situation and education level.

and we have added this in the methods section

A targeted sampling strategy was used to capture a range of gender, age, body mass index, living situations, education level and diabetes type (Table 1).

The Results section is presented in too narrative form. The sentences: "Many of participants referred…." or "Some expressed" must be avoided since no information about the numerical relevance of the phenomena is provided. For the same reason, the individual statements have to be eliminated. A table reporting the percentage distribution of the answers could help in analyzing the main results of the interview.

Thank you for this comment. We appreciate that if utilising a quantitative approach this would be seen as imprecise. However, with reflexive thematic analysis in qualitative research, the quantity of each opinion is not viewed as important but rather, that the theme captures something important in relation to the research question. There is no hard and fast answer to the question of what proportion of the data needs to display evidence to the theme. Researcher judgement is necessary to determine what a theme is, and that this is flexible. The keyness of a theme is not necessarily dependent on quantifiable measures - but rather whether it captures something important in relation to the overall research question [2].

The use of quotes to support themes are consistent with qualitative approach and help guide the reader and ensure trustworthiness of the data, as such they are an important part of the results section and we politely disagree, that these should not be removed. 

The Introduction is too long and several concepts and comments are present in both Introduction and Discussion (lines 52-55 and 392-397), or discussed in the Results.

Thank you and we agree there was some repetition and that the introduction was too long. Large sections of text have been removed to rectify this. 

Even if the patients number is very small, the correlation between the perception of food and some parameters (for example BMI class, or Diabetes type or duration) should be evaluated, this improving the impact of the study.

Thank you we agree that this would be an interesting question to answer. However this was not the aim of the current study, and performing an analysis such as you have suggested is not consistent with the qualitative method we have utilised. 

  1. Terry G, Hayfield N. Essentials of Thematic Analysis. Washington, DC 20002: American Psychological Association; 2021 31/05/2022.
  2. Braun V, Clarke V. Using thematic analysis in psychology. Qualitative Research in Psychology. 2006;3(2):77-101.
  3. Collins R, Burrows T, Donnelly H, Tehan PE. Macronutrient and micronutrient intake of individuals with diabetic foot ulceration: A short report. J Hum Nutr Diet. 2021.
  4. Armstrong DG MJ, Molina M, Molnar JA. Nutrition interventions in adults with diabetic foot ulcers. Guideline Central. 2021.

5. Terry G, Hayfield N, Clarke V, Braun V. Thematic Analysis. In: Willig C, Stainton RW, editors. The sage handbook of qualitative research in psychology SAGE Publications; 2017. p. 18.

Round 2

Reviewer 3 Report

Not applicable. See comments for Editor